# Diagnostic accuracy of the Copenhagen Index in ovarian malignancy: A meta-analysis

**Huiling Liu**[�], **Shouye Ma**[ID]*[☯], **Xiaohong Chen, Huifang Wu, Rongrong Wang, Mengmeng Du, Xiazi Nie**

Obstetrics and Gynecology Department, Gansu Provincial Hospital, Lanzhou, Gansu, China

☯ These authors contributed equally to this work.
* msymashouye@163.com

## Abstract

### Objective

To assess the diagnostic value of the Copenhagen index for ovarian malignancy.

### Methods

PubMed, Web of Science, the Cochrane Library, Embase, CBM, CNKI, and WanFang databases were searched throughout June 2021. Statistical analyses were performed using Stata 12, Meta-DiSc, and RevMan 5.3. The pooled sensitivity, specificity, and diagnostic odds ratio were calculated, the summary receiver operating characteristic curve was drawn, and the area under the curve was calculated.

### Results

Ten articles, including 11 studies with a total of 5266 patients, were included. The pooled sensitivity, specificity, and diagnostic odds ratio were 0.82 [95% CI (0.80–0.83)], 0.88 [95% CI (0.87–0.89)], and 57.31 [95% CI (32.84–100.02)], respectively. The area under the summary receiver operating characteristics curve and the Q index were 0.9545 and 0.8966, respectively.

### Conclusion

Our systematic review shows that the sensitivity and specificity of the Copenhagen index are high enough for it to be used in a clinical setting to provide accurate ovarian cancer diagnosis without considering menopausal status.

## Introduction

Ovarian cancer has the highest mortality rate among gynecological malignant tumors and is the fifth most common cause of cancer-related deaths among women, with more than 150,000 yearly deaths worldwide [1]. The occurrence of ovarian cancer is insidious in nature; 70–75% of patients are in an advanced stage at the first hospital visit, and the 5-year survival rate of

**Data Availability Statement:** All relevant data are within the paper and its Supporting Information files.

**Funding:** This work was supported by the National Natural Science Foundation of China (grant no.

81560426), the Key Discipline Project of Gansu Provincial Hospital (grant no. 20GSSY1-4).The funders had no role in study design, data collection and analysis, decision to publish, or preparation of the manuscript.

**Competing interests:** The authors have declared that no competing interests exist.

advanced patients is only about 20–30%. However, the 5-year survival rate of early-stage ovarian cancer patients is high, and the 5-year survival rate of stage I patients can be as high as 90–95% [2, 3]. Studies have shown that although the survival rate of patients with ovarian cancer in developed countries is increasing every year, morbidity and mortality rates are still high [4, 5]. Therefore, improving the diagnosis rate of early-stage ovarian cancer and ensuring appropriate management of this disease can successfully reduce the mortality rate, which is crucial for improving the prognosis [6].

At present, the main screening methods for ovarian cancer are color ultrasound imaging and detection of tumor markers. Ultrasound examination is subjective, while tumor markers have the advantages of objective results, low cost, and easy popularization. Carbohydrate antigen 125 (CA125) has recently been recognized as a method for screening and evaluating ovarian cancer, but it is prone to false positives and has low specificity and sensitivity for the detection of ovarian cancer [7]. In contrast, human epididymis protein (HE4), a new marker for the diagnosis of ovarian cancer, particularly early-stage ovarian cancer, has high specificity and is present in very high levels in the serum of patients with ovarian cancer [7]. It was found that the combination of the two tumor markers can significantly improve the sensitivity and specificity of detection, and can better predict ovarian malignant tumors than a single index [8].

In recent years, to improve the accuracy of ovarian cancer diagnosis and to find more effective predictors, scholars have developed the risk of malignancy index (RMI), risk of ovarian malignancy algorithm (ROMA), and the Copenhagen Index (CPH-I) [9–11]. CPH-I, a predictive model based on serum CA125, and HE4 levels, and age of the patient (Karlsen et al., 2015 [11]), has been confirmed to have a similar ability to distinguish epithelial ovarian cancer from benign ovarian tumors as that of ROMA and RMI. Yoshida et al. [12] further confirmed that an important improvement in the CPH-I was to use age instead of menstrual status or ultrasonic score, resulting in broader application prospects. The purpose of our meta-analysis is to confirm the accuracy of CPH-I in predicting ovarian malignant tumors.

## Methods

### Search strategy

We systematically searched PubMed, Web of Science, the Cochrane Library, Embase, CBM, CNKI, and WanFang databases for studies on the differential diagnosis of benign and malignant ovarian cancer, which were published from inception to June 2021. Simultaneously, the references in the literature were traced back to supplement the relevant findings by online searches using combinations of subject headings or other keywords, such as: "ovarian tumor" OR "ovarian cancer" OR "ovarian carcinoma" OR "ovarian malignancy" OR "ovarian neoplasms" AND "Copenhagen Index,CPH-I". The PubMed database was searched as follows (Box 1):

### Box 1

#1 (((((ovarian tumor[Title/Abstract]) OR (ovarian cancer[Title/Abstract])) OR (ovarian carcinoma[Title/Abstract])) OR (ovarian malignancy[Title/Abstract])) OR (ovarian neoplasms [Title/Abstract])
　#2 Copenhagen Index[Title/Abstract]
　#3 #1 AND #2

### Study selection

It was ensured that the included studies met the following criteria: (1) The study focused on the value of CPH-I in the differential diagnosis of benign and malignant ovarian tumors (2)

The subjects were patients with "adnexal masses" (3) None of the patients received radiotherapy, chemotherapy, or tumor drug treatment before surgery (4)The gold standard was the postoperative pathological information (5) The study was able to extract complete four-grid data (true positive value, false positive value, false negative value, true negative value). The following studies were excluded: (1) Conference abstracts, letters, reviews, case reports, commentaries, and expert opinions (2) Studies with incomplete data, poor quality, or lack of access to the full text.

### Quality assessment and data extraction

The quality of the study was independently assessed by two investigators according to the Quality Assessment of Diagnostic Accuracy Studies (QUADAS-2) [13]. Two investigators independently extracted the data that met our inclusion and exclusion criteria. Discrepancies were resolved by consensus. We extracted the following information: name of the first author, year of publication, country, research type, sample size, and main parameters of the diagnostic test (true positive, false positive, true negative, false negative).

### Statistical analyses

Meta-DiSc1.4, RevMan 5.3 and Stata 12 software were used for statistical analyses. First, Spearman correlation analysis was used to check whether there was heterogeneity caused by the threshold effect, and combined with the $I^2$ value to quantitatively judge the size of heterogeneity [14]; the fixed-effect model was used for merger analysis if $I^2 \leq 50\%$ and the random-effect model was used if $I^2 > 50\%$. The pooled sensitivity, specificity and diagnostic odds ratio (DOR) included in the study were calculated. A summary receiver operating characteristic (SROC) curve was constructed, and the area under the curve (AUC) and Q index were calculated. The sensitivity analysis showed that the difference was statistically significant when $p < 0.05$. Finally, Deeks' test was used to statistically assess publication bias, and publication bias was considered to exist when $p < 0.05$ [15].

## Results

According to the inclusion and exclusion criteria, ten articles [11, 12, 16–23], including 11 studies and 5266 women, were included in the meta-analysis after excluding duplicate citations, reviews, non-diagnostic trial studies, and articles that did not meet the inclusion criteria (Fig 1). Their basic characteristics are summarized in Table 1. The quality evaluation of the included articles is shown in Fig 2: six articles [11, 16, 17, 21–23] met the QUADAS-2 standard, one article [19] had a high risk of bias in the index test, one article [20] was not clear about the risk of bias, and two articles [12, 18] showed an unclear risk of bias in the flow and timing.

### Threshold effect analysis

Due to the difference in the cutoff value, there were differences in sensitivity, specificity, and DOR, resulting in the generation of a threshold effect. Therefore, the existence of a threshold effect was tested first. The SROC curves drawn using Meta-Disc software (Fig 3) indicated that the position of none of the studies in the curves showed a typical shoulder-arm distribution. These results, along with the Spearman correlation coefficient of -0.073 (p = 0.832) indicated that there was no threshold effect in the study.

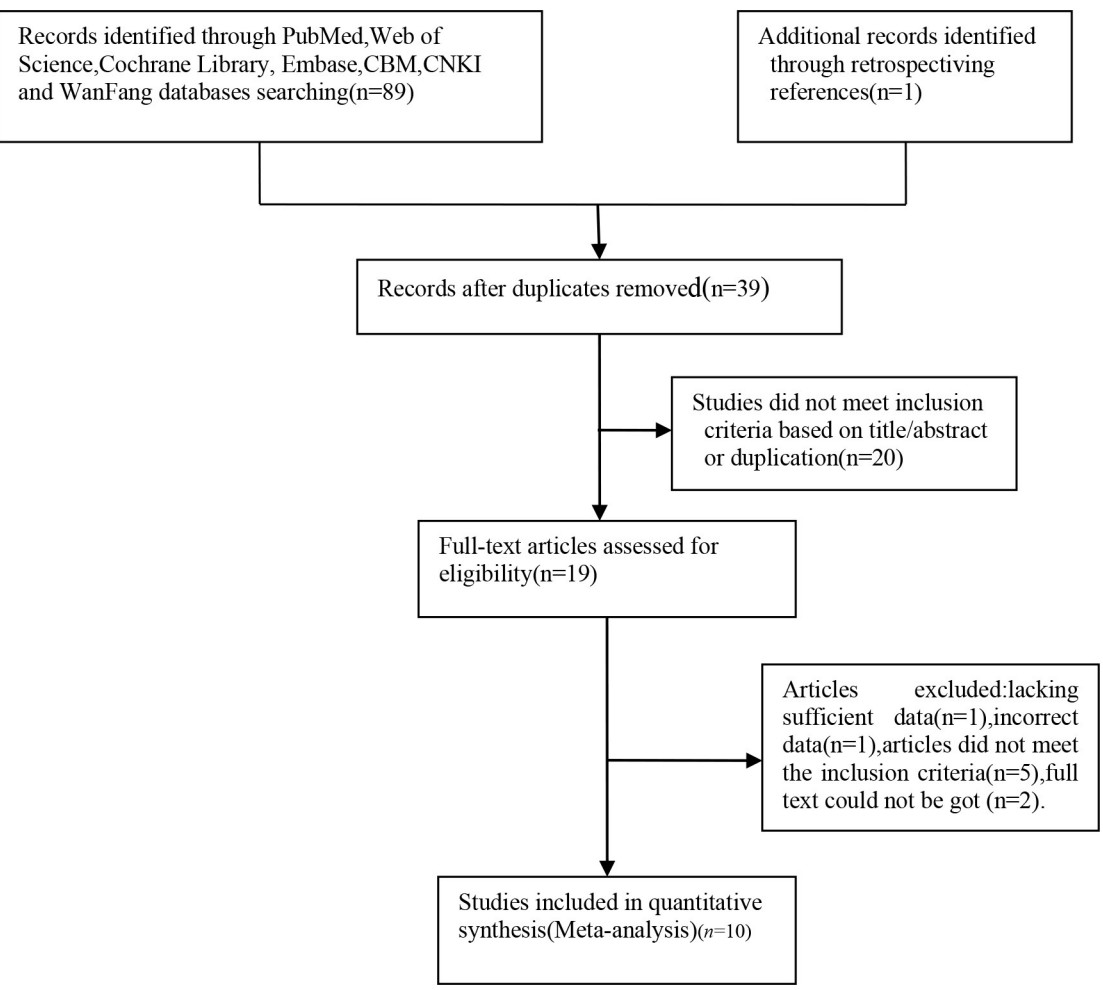

**Fig 1. Flow diagram of the study selection process.**

Table 1. Characteristics of the included studies.

| Study | Country | Research type | No. of P. | TP | FP | FN | TN | Cutoff point |
|---|---|---|---|---|---|---|---|---|
| Chen 2018 [16] | China | Retro. | 180 | 75 | 1 | 16 | 88 | 7.0% |
| Gong 2017 [17] | China | Retro. | 208 | 91 | 4 | 9 | 104 | 7.0% |
| Gong 2019 [18] | China | Retro. | 719 | 120 | 28 | 68 | 503 | 7.0% |
| Karlsen 2015 [11][a] | Denmark | Pro. | 1055 | 234 | 175 | 12 | 634 | 7.0% |
| Karlsen 2015 [11][b] | Eight international studies | Pro. | 1610 | 451 | 123 | 99 | 937 | 7.0% |
| Lu 2020 [19] | China | Retro. | 143 | 36 | 4 | 8 | 95 | 12.42% |
| Ma 2018 [20] | China | Retro. | 120 | 60 | 7 | 0 | 103 | - |
| Minar 2018 [21] | Czech Republic | Retro. | 267 | 108 | 17 | 49 | 93 | 7.0% |
| Yoshida 2016 [12] | Brazil | Pro. | 384 | 117 | 35 | 43 | 189 | 7.0% |
| Yu 2019 [22] | China | Retro. | 515 | 176 | 22 | 29 | 288 | 7.0% |
| Yuan 2020 [23] | China | Retro. | 65 | 41 | 1 | 4 | 19 | 7.0% |

No. of P., number of patients; TP, true positive value; FP, false positive value; FN, false negative value; TN, true negative value; Retro., retrospective study; Pro., prospective study; -, unclear.

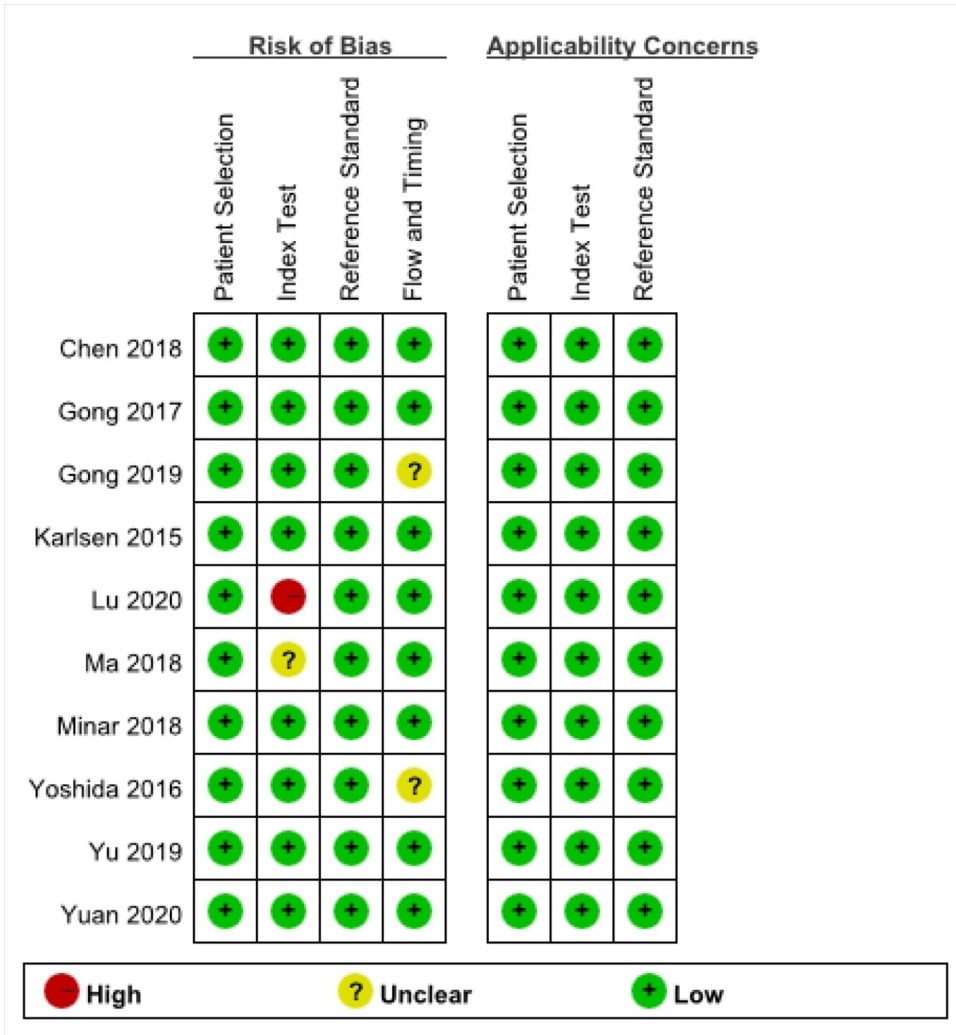

**Fig 2. Quality evaluation of included articles.**

## Meta-analysis of combined effect quantity

Eleven studies reported the value of CPH-I in the differential diagnosis of benign and malignant ovarian tumors. The sensitivity, specificity, and DOR were tested using a heterogeneity test, and their chi-square values were 92.4%, 92.7%, and 84.6%, respectively (p < 0.001). It was suggested that sensitivity, specificity, and DOR heterogeneity were high for CPH-I in the differential diagnosis of benign and malignant ovarian tumors, therefore, the random-effect model was used for combination analysis. The results showed that the combined sensitivity is 0.82 [95% CI (0.80–0.83)], the combined specificity is 0.88 [95% CI (0.87–0.89)], the combined DOR is 57.31 [95% CI (32.84–100.02)], and the SROC curve (AUC) and Q index are 0.9545 and 0.8966, respectively. Forest maps of sensitivity, specificity, DOR and SROC curves are shown in Figs 3–6.

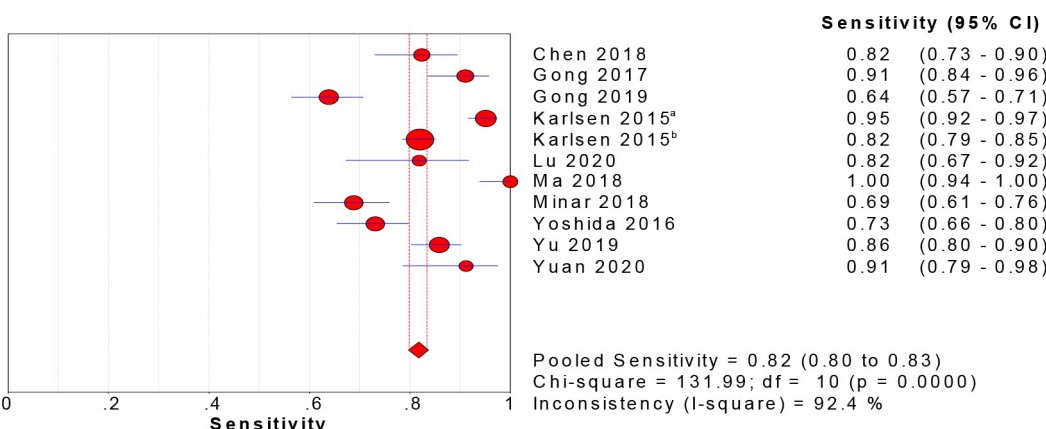

**Fig 3. SROC curve of CPH-I in differential diagnosis of benign and malignant ovarian tumors.**

**Fig 4. Sensitivity forest map of CPH-I in predicting ovarian malignant tumor.**

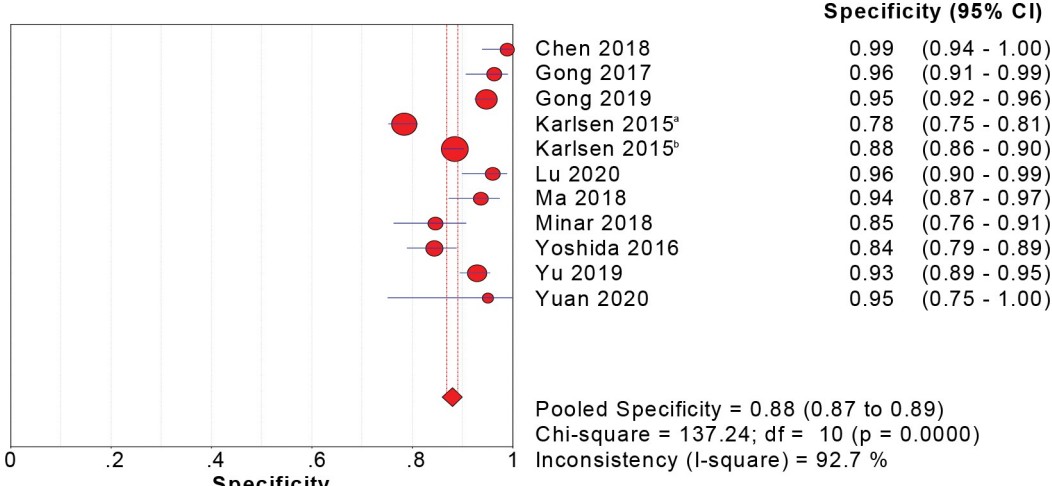

**Fig 5. Specificity forest map of CPH-I in predicting ovarian malignant tumor.**

### Sensitivity analysis

A sensitivity analysis omitting one study at a time and calculating the pooled DORs of the remaining studies showed that there were no significant changes in the DOR and I² (Table 2), suggesting that the results of this analysis do not rely on a single study, and the conclusions were reliable.

### Publication bias

Deeks' test was used to assess the publication bias in the meta-analysis. There is no indication for publication bias as per Deeks' test (p = 0.111) (Fig 7).

## Discussion

To date, no tumor marker can accurately detect early-stage ovarian cancer [24], which results in patients losing their crucial treatment time. At present, CA125 is still the preferred marker for the clinical diagnosis of ovarian cancer, but due to its low specificity, the United States Preventive Services Task Force does not recommend its use for ovarian cancer screening in

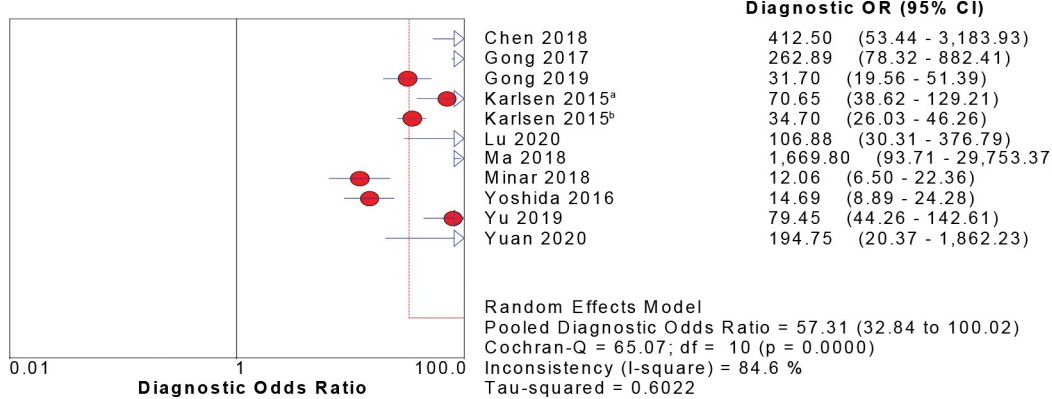

**Fig 6. DOR forest map of CPH-I in predicting ovarian malignant tumor.**

**Table 2. Sensitivity analyses.**

| Groups | Studies(n) | DOR(95%CI) | Heterogeneity test | |
|---|---|---|---|---|
| | | | P | I²(%) |
| All studies | 11 | 57.31(32.84~100.02) | 0.000 | 84.6 |
| Omitting Chen 2018 [16] | 10 | 51.22(29.48~89.00) | 0.000 | 84.8 |
| Omitting Gong 2017 [17] | 10 | 48.55(28.11~83.85) | 0.000 | 83.5 |
| Omitting Gong 2019 [18] | 10 | 65.37(34.14~125.15) | 0.000 | 86.1 |
| Omitting Karlsen 2015 [11]ᵃ | 10 | 56.45(30.84~103.34) | 0.000 | 84.9 |
| Omitting Karlsen 2015 [11]ᵇ | 10 | 68.11(33.34~139.13) | 0.000 | 86.2 |
| Omitting Lu 2020 [19] | 10 | 54.30(30.40~96.98) | 0.000 | 85.5 |
| Omitting Ma 2018 [20] | 10 | 50.90(29.67~87.33) | 0.000 | 84.4 |
| Omitting Minar 2018 [21] | 10 | 68.54(38.89~120.77) | 0.000 | 82.7 |
| Omitting Yoshida 2016 [12] | 10 | 68.07(38.22~121.25) | 0.000 | 82.3 |
| Omitting Yu 2019 [22] | 10 | 55.24(30.35~100.53) | 0.000 | 84.4 |
| Omitting Yuan 2020 [23] | 10 | 54.22(30.78~95.52) | 0.000 | 85.7 |

a:Denmark

b: Eight international studies

asymptomatic women with an uncertain genetic risk [25]. Kirehoff et al. [26] found that the HE4 gene, identified in human distal epididymal epithelial cells, was expressed at high levels in ovarian cancer tissues, but was not expressed, or expressed at low levels in benign ovarian tissues and normal tissues suggesting that it had good specificity and compensated for the

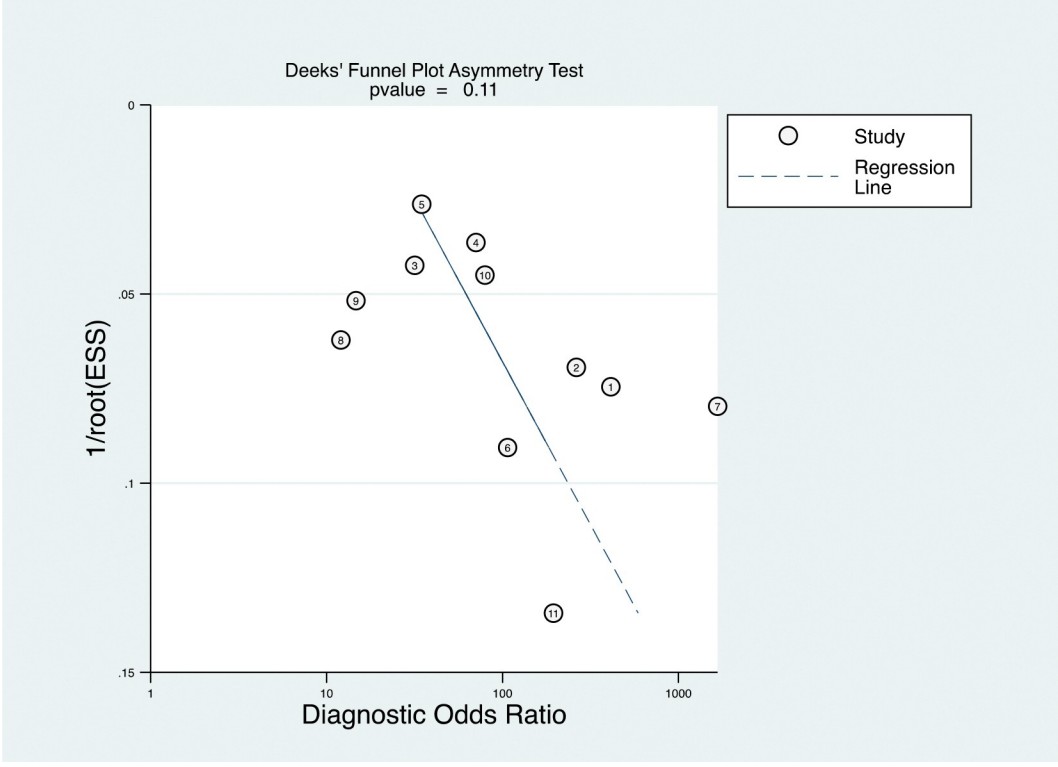

**Fig 7. Funnel plot of CPH-I.**

drawbacks of CA125. These findings indicated that the combination of these two markers could improve the accuracy of diagnosis. In order to further improve the accuracy of diagnosis, some scholars have established mathematical models based on serological markers, ultrasound examination and menopausal status: RMI and ROMA. RMI incorporated serum CA125 levels, ultrasound score, and menopausal status in the formula to predict the risk of ovarian malignancies, whereas ROMA, combined with serum CA125 and HE4 could predict the risk of ovarian malignant tumors before and after menopause, according to the menopausal status. Due to the strong subjectivity of ultrasound score and the inconsistent definition of menopause by different researchers, the results may be biased. Therefore, it is still controversial whether ROMA and RMI have a better ability for the differential diagnosis of benign and malignant ovarian tumors than a single index [27–29]. Compared with menopause, age is clear and is not affected by external factors. One study [30] has shown that age is the most important independent risk factor for ovarian cancer. Based on the above theory, Karlsen et al. [11] conducted a prospective multicenter international study in 2015. According to the level of serum CA125, and HE4, and age of patients, CPH-I was proposed for the differential diagnosis of benign and malignant ovarian tumors, which was as effective as ROMA and superior to RMI. Our meta-analysis results showed that the combined sensitivity and specificity of CPH-I in predicting ovarian malignant tumors were 82% and 88%, respectively, indicating that the failed diagnosis rate was 18% and the misdiagnosis rate was 12%.SROC curve is an effective method for evaluating diagnostic tests comprehensively and accurately; the higher the value of the curve, the higher the diagnostic value. The AUC of CPH-I in our study was 0.9545, indicating that the accuracy of diagnosis was very high. The Q index is the corresponding value of the point closest to the upper left corner of the SROC curve; at this point, the sensitivity and specificity are equal. Higher the Q index, higher is the accuracy of the diagnostic test. The Q index of our study was 0.8966, which together with the area under the SROC, shows that the accuracy of CPH-I in the diagnosis of ovarian malignant tumors is very good.

Therefore, CPH-I combines the three parameters of serum CA125 level, serum HE4 level, and patient age, and satisfactorily predicts the benign and malignant ovarian tumors. In addition, Yoshida et al. [12] conducted a study on 384 patients with ovarian tumors and found that the sensitivity of CPH-I in predicting ovarian malignant tumors was 73.1%, the specificity was 84.4%, and the AUC was 0.84. Consistently, Minar et al. [21] also showed that the sensitivity, specificity, and AUC of CPH-I in the diagnosis of ovarian malignant tumors were 69%, 85%, and 0.81, respectively. Gong et al. [18] found that the sensitivity and specificity of CPH-I in the differential diagnosis of ovarian malignant tumors were 63.8% and 94.7%, respectively. Notably, borderline tumors, non-epithelial ovarian tumors, and metastatic tumors were not specifically excluded in these studies, thereby mimicking the clinical situation.

In our study, the literature was screened strictly, and the included studies were of high quality. Our meta-analysis found that the sensitivity and specificity of CPH-I in the differential diagnosis of benign and malignant ovarian tumors were high, and the diagnostic efficacy of CPH-I was not affected by menopausal status. Age was relatively easy to obtain and popularize in clinics, compared with menopausal status. Therefore, CPH-I can be considered as a simple substitute for ROMA in clinics to predict benign and malignant ovarian tumors. It can be used to screen patients with suspected ovarian cancer, formulate an effective and reasonable diagnosis and treatment plan, or provide a timely referral. There were also limitations to this meta-analysis. First, although the literature screening criteria of this meta-analysis were relatively strict, the heterogeneity among studies was still high, which may be due to differences in sample size, study design, and critical values of each study. Second, the potential for publication bias is present in this article, as small studies with null results are often unpublished, and there may also be bias in outcome reporting. In conclusion, despite the above limitations, our meta-

analysis supports the fact that the diagnostic efficacy of CPH-I is high in predicting ovarian malignant tumors without considering menopausal status.

The conclusion of our study has high clinical value for the early diagnosis of ovarian cancer and is worth popularizing, however, further prospective evaluation needs to be conducted in the future.

## Supporting information

**S1 File. The appropriate data.**
(XLSX)

**S1 Checklist. PRISMA 2020 checklist.**
(DOCX)

## Author Contributions

**Conceptualization:** Shouye Ma.

**Data curation:** Shouye Ma, Rongrong Wang, Mengmeng Du, Xiazi Nie.

**Formal analysis:** Xiaohong Chen, Huifang Wu.

**Methodology:** Xiaohong Chen, Huifang Wu.

**Project administration:** Shouye Ma.

**Software:** Rongrong Wang, Mengmeng Du, Xiazi Nie.

**Supervision:** Huiling Liu, Shouye Ma.

**Validation:** Huiling Liu, Shouye Ma.

**Writing – original draft:** Huiling Liu, Shouye Ma.

**Writing – review & editing:** Huiling Liu, Shouye Ma.

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
