## [Decision Letter · Decision Letter 0]

15 Aug 2022

PONE-D-21-26313Diagnostic accuracy of the Copenhagen Index in ovarian malignancy: a meta-analysisPLOS ONE

Dear Dr. Ma,

Thank you for submitting your manuscript to PLOS ONE. After careful consideration, we feel that it has merit but does not fully meet PLOS ONE’s publication criteria as it currently stands. Therefore, we invite you to submit a revised version of the manuscript that addresses the points raised during the review process.

We look forward to receiving your revised manuscript.

Kind regards,

Federico Ferrari, MD, PhD

Academic Editor

PLOS ONE

Journal Requirements:

2. Thank you for providing the date(s) when patient medical information was initially recorded. Please also include the date(s) on which your research team accessed the databases/records to obtain the retrospective data used in your study.

6. Please upload a new copy of Figure 2 as the detail is not clear. Please follow the link for more information: https://blogs.plos.org/plos/2019/06/looking-good-tips-for-creating-your-plos-figures-graphics/" https://blogs.plos.org/plos/2019/06/looking-good-tips-for-creating-your-plos-figures-graphics/

Reviewers' comments:

Reviewer's Responses to Questions

**Comments to the Author**

1. Is the manuscript technically sound, and do the data support the conclusions?

Reviewer #1: Yes

Reviewer #2: Yes

2. Has the statistical analysis been performed appropriately and rigorously? 

Reviewer #1: Yes

Reviewer #2: Yes

3. Have the authors made all data underlying the findings in their manuscript fully available?

Reviewer #1: Yes

Reviewer #2: Yes

4. Is the manuscript presented in an intelligible fashion and written in standard English?

Reviewer #1: No

Reviewer #2: Yes

5. Review Comments to the Author

Reviewer #1: • Line 70: During the last 5 years, the international literature on Copenhagen Index (CPH-I) has been increased. Authors are recommended to recheck PubMed for papers (up to June 2021, as stated) for further eligible studies, i.e. check the article PMID: 33994145 for eligibility for inclusion.

• Lines 197-199: the authors are recommended to provide more discussion on the value of CPH-I in non-epithelial ovarian /adnexal tumors

• Lines 210-211: It’s recommended to place the limitations of the study at the end of Discussions, not within the Conclusion.

• Line 215: “…because small studies with null results tend to remain unpublished,…”: this comment/judgement is too subjective and should be re-formulated.

• Check English syntax and grammar

Reviewer #2: This manuscript of Liu et colleagues (entitled Diagnostic accuracy of the Copenhagen Index in ovarian malignancy: a meta-analysis) addresses an interesting topic. The Copenhagen Index algorithm (CPH-I), published by Karlsen et al in 2015, considered three elements: measurements of CA125, HE4, and patient’s age. The serum CA125 is strictly related to the stage of the disease: higher serum levels are present in 80%–90% of women with stage III–IV disease, but approximately in 50% of stage I disease. HE4 has higher specificity and lower false-positivity than CA125, but could be associated with others disease (as lung and endometrial cancer, renal disease or endometriosis). In comparison to ROMA score, CPH-I doesn’t consider the menopausal status but only the age of the patient. In fact, menopause diagnosis is not standardized, thus HE4 serum levels increase with age regardless of menopause. CPH-I was as effective as ROMA and superior to RMI. The gold standard of different diagnosis is considered gynecological imaging like transvaginal ultrasound. However this exam depends on skill level of the operator. Thus, in non-specialized centers auxiliary methods such as CPH-I and ROMA may be useful to refer selected patients with malignant disease to a specialized cancer center. I appreciate the novelty of the paper, and the design of the study; all important data are reported and figures are appropriate and useful. Method section is complete. Statistical analysis is well conducted and adequate to highlight the potential clinical role and technical reliability . I advice to update the background section with a more complete description of Copenhagen Index to provide a complete overview. In the Results I suggest to introduce a separate “Quality assessment” section. I also suggest to considered this paper DOI: 10.1016/j.ygyno.2021.05.001 about the evaluation of ROMA vs CPH-I, which showed a median values of CPH-I and ROMA lower than those of some other studies. In conclusion, discerning a benign from a malignant tumor before surgery is crucial to the correct referral the woman, considering that survival of ovarian cancer patients is highly dependent on the primary surgical and oncological treatment. In non-specialized centers, triaging methods such as CPH-I and ROMA can be more useful to select patients for referral to highly specialized gynaecological cancer centers. In clinical practice the CPH-I might be considered as an alternative to ROMA, with a similar accuracy but simpler, easy to collect and more objective.

6. PLOS authors have the option to publish the peer review history of their article (what does this mean?). If published, this will include your full peer review and any attached files.

Reviewer #1: No

Reviewer #2: **Yes: **Francesca Cisotto

---

## [Author Response · Author response to Decision Letter 0]

28 Sep 2022

Dear editor and reviewers,

Re: “Diagnostic accuracy of the Copenhagen Index in ovarian malignancy: a meta-analysis " (PONE-D-21-26313)

Thanks to the editor and reviewers for the insightful comments and the valuable suggestions for revision of our MS. We have made changes in accordance with the suggestions from the editor and the reviewers in our revised MS. All of the modifications are highlighted in red in the text. The point by point response to the editor and reviewers’ comments was attached in the following part. We hope the changes incorporated would make the manuscript acceptable for publication in the Journal of PLOS ONE.

Editor Comments:

Question 1: Please ensure that your manuscript meets PLOS ONE's style requirements, including those for file naming.

Response: Thanks for your question. We have examined the MS and ensured that our manuscript meets PLOS ONE's style requirements. 

Question 2 : Thank you for providing the date(s) when patient medical information was initially recorded. Please also include the date(s) on which your research team accessed the databases/records to obtain the retrospective data used in your study.

Response: Thanks for your question. We have provided the date on which our research team accessed the databases/records to obtain the retrospective data used in our study in Search strategy section.

Question 3 : We note that the grant information you provided in the ‘Funding Information’ and ‘Financial Disclosure’ sections do not match. When you resubmit, please ensure that you provide the correct grant numbers for the awards you received for your study in the ‘Funding Information’ section.

Response: Thanks for your question. We have provided the correct grant numbers for the awards.

Response: Thanks for your question. The appropriate data supporting the conclusions of this article is included within the Supporting information:S1 file.

Reviewer 1:

Question 1 : Line 70: During the last 5 years, the international literature on Copenhagen Index (CPH-I) has been increased. Authors are recommended to recheck PubMed for papers (up to June 2021, as stated) for further eligible studies, i.e. check the article PMID: 33994145 for eligibility for inclusion.

Response: Thanks for your question insightful comment. We have checked the article PMID: 33994145 for eligibility for inclusion, which did not meet our inclusion criteria.

Question 2: Lines 197-199: the authors are recommended to provide more discussion on the value of CPH-I in non-epithelial ovarian /adnexal tumors

Response: Thanks for your question. We have provided more discussion on the value of CPH-I in non-epithelial ovarian /adnexal tumors.

Question 3: Lines 210-211: It’s recommended to place the limitations of the study at the end of Discussions, not within the Conclusion.

Response: Thanks for your question. We have placed the limitations of the study at the end of Discussions.

Question 4: • Line 215: “…because small studies with null results tend to remain unpublished,…”: this comment/judgement is too subjective and should be re-formulated.

• Check English syntax and grammar

Response: Thanks for your question. We have modified the English syntax and grammar: The potential for publication bias is present in this article, as small studies with null results are often unpublished, and there may also be bias in outcome reporting.

Reviewer 2:

This manuscript of Liu et colleagues (entitled Diagnostic accuracy of the Copenhagen Index in ovarian malignancy: a meta-analysis) addresses an interesting topic. The Copenhagen Index algorithm (CPH-I), published by Karlsen et al in 2015, considered three elements: measurements of CA125, HE4, and patient’s age. The serum CA125 is strictly related to the stage of the disease: higher serum levels are present in 80%–90% of women with stage III–IV disease, but approximately in 50% of stage I disease. HE4 has higher specificity and lower false-positivity than CA125, but could be associated with others disease (as lung and endometrial cancer, renal disease or endometriosis). In comparison to ROMA score, CPH-I doesn’t consider the menopausal status but only the age of the patient. In fact, menopause diagnosis is not standardized, thus HE4 serum levels increase with age regardless of menopause. CPH-I was as effective as ROMA and superior to RMI. The gold standard of different diagnosis is considered gynecological imaging like transvaginal ultrasound. However this exam depends on skill level of the operator. Thus, in non-specialized centers auxiliary methods such as CPH-I and ROMA may be useful to refer selected patients with malignant disease to a specialized cancer center. I appreciate the novelty of the paper, and the design of the study; all important data are reported and figures are appropriate and useful. Method section is complete. Statistical analysis is well conducted and adequate to highlight the potential clinical role and technical reliability . I advice to update the background section with a more complete description of Copenhagen Index to provide a complete overview. In the Results I suggest to introduce a separate “Quality assessment” section. I also suggest to considered this paper DOI: 10.1016/j.ygyno.2021.05.001 about the evaluation of ROMA vs CPH-I, which showed a median values of CPH-I and ROMA lower than those of some other studies. In conclusion, discerning a benign from a malignant tumor before surgery is crucial to the correct referral the woman, considering that survival of ovarian cancer patients is highly dependent on the primary surgical and oncological treatment. In non-specialized centers, triaging methods such as CPH-I and ROMA can be more useful to select patients for referral to highly specialized gynaecological cancer centers. In clinical practice the CPH-I might be considered as an alternative to ROMA, with a similar accuracy but simpler, easy to collect and more objective.

Response: Thanks for your question and insightful comment. We have updated the background section. CPH-I, a new tumor index calculated by HE4, CA125, and age rather than menopausal status with different definitions, was thought to be easier to obtain than ROMA clinically. Researches have shown that CPH-I and ROMA have similar discriminatory performance in benign lesions and malignant ovarian tumors

Thank you for your kind consideration. 

Sincerely,

Dr. Xiaoling Gao, professor, Director

The Institute of Clinical Research and Translational Medicine

Gansu Provincial Hospital

No. 204 Donggang West Road, Chengguan District, Lanzhou, 730000, China.

E-mail addresses: gaoxl008@hotmail.com

Tel: +86-931-8281222; 

Fax: +86-931-8281222.

---

## [Decision Letter · Decision Letter 1]

6 Mar 2023

PONE-D-21-26313R1Diagnostic accuracy of the Copenhagen Index in ovarian malignancy: a meta-analysisPLOS ONE

Dear Dr. Ma,

Thank you for submitting your manuscript to PLOS ONE. After careful consideration, we feel that it has merit but does not fully meet PLOS ONE’s publication criteria as it currently stands. Therefore, we invite you to submit a revised version of the manuscript that addresses the points raised during the review process.

We look forward to receiving your revised manuscript.

Kind regards,

Federico Ferrari, MD, PhD

Academic Editor

PLOS ONE

Journal Requirements:

Reviewers' comments:

Reviewer's Responses to Questions

**Comments to the Author**

1. If the authors have adequately addressed your comments raised in a previous round of review and you feel that this manuscript is now acceptable for publication, you may indicate that here to bypass the “Comments to the Author” section, enter your conflict of interest statement in the “Confidential to Editor” section, and submit your "Accept" recommendation.

Reviewer #1: All comments have been addressed

Reviewer #3: (No Response)

2. Is the manuscript technically sound, and do the data support the conclusions?

Reviewer #1: Yes

Reviewer #3: Yes

3. Has the statistical analysis been performed appropriately and rigorously? 

Reviewer #1: Yes

Reviewer #3: Yes

4. Have the authors made all data underlying the findings in their manuscript fully available?

Reviewer #1: Yes

Reviewer #3: Yes

5. Is the manuscript presented in an intelligible fashion and written in standard English?

Reviewer #1: Yes

Reviewer #3: Yes

6. Review Comments to the Author

Reviewer #1: The revised manuscript already addressed all reviewer's comments and can now be considered for acceptance.

Reviewer #3: Please, modify "maging" with imaging in 44 line.

I reccomend to add a reference for the sentence: "Carbohydrate antigen 125 (CA125) has recently been recognized as a method for screening and evaluating ovarian cancer, but it is prone to false positives and has low specificity and sensitivity for the detection of ovarian cancer".

7. PLOS authors have the option to publish the peer review history of their article (what does this mean?). If published, this will include your full peer review and any attached files.

Reviewer #1: No

Reviewer #3: **Yes: **Francesco Marasciulo

---

## [Author Response · Author response to Decision Letter 1]

20 Apr 2023

Dear editor and reviewers,

Re: “Diagnostic accuracy of the Copenhagen Index in ovarian malignancy: a meta-analysis " (PONE-D-21-26313R1)

Thanks to the editor and reviewers for the insightful comments and the valuable suggestions for revision of our MS. We have made changes in accordance with the suggestions from the editor and the reviewers in our revised MS. All of the modifications are highlighted in red in the text. The point by point response to the editor and reviewers’ comments was attached in the following part. We hope the changes incorporated would make the manuscript acceptable for publication in the Journal of PLOS ONE.

Editor Comments:

Question 1: Please review your reference list to ensure that it is complete and correct. If you have cited papers that have been retracted, please include the rationale for doing so in the manuscript text, or remove these references and replace them with relevant current references. Any changes to the reference list should be mentioned in the rebuttal letter that accompanies your revised manuscript. If you need to cite a retracted article, indicate the article’s retracted status in the References list and also include a citation and full reference for the retraction notice.

Response: Thanks for your question. We have examined the reference list that it is correct. In addition, we have added a reference basing on the comment of Reviewer 3 , which was highlighted in red in the reference list. 

Reviewer 3:

Question 1 :Please, modify "maging" with imaging in 44 line.

I reccomend to add a reference for the sentence: "Carbohydrate antigen 125 (CA125) has recently been recognized as a method for screening and evaluating ovarian cancer, but it is prone to false positives and has low specificity and sensitivity for the detection of ovarian cancer".

Response: Thanks for your question.We have modified "maging" with imaging in 44 line. We have added a reference for the sentence: "Carbohydrate antigen 125 (CA125) has recently been recognized as a method for screening and evaluating ovarian cancer, but it is prone to false positives and has low specificity and sensitivity for the detection of ovarian cancer".It was highlighted in red in the reference list. 

Thank you for your kind consideration. 

Sincerely,

Ms Shouye Ma, MD 

Obstetrics and Gynecology Department 

Gansu Provincial Hospital

No. 204 Donggang West Road, Chengguan District, Lanzhou, 730000, China.

E-mail addresses: msymashouye@163.com

Tel: +86-931-8281060; 

Fax: +86-931-8281060.

---

## [Editor Report · Decision Letter 2]

22 May 2023

Diagnostic accuracy of the Copenhagen Index in ovarian malignancy: a meta-analysis

PONE-D-21-26313R2

Dear Dr. Ma,

We’re pleased to inform you that your manuscript has been judged scientifically suitable for publication and will be formally accepted for publication once it meets all outstanding technical requirements.

Kind regards,

Federico Ferrari, MD, PhD

Academic Editor

PLOS ONE
---

## [Editor Report · Acceptance letter]

4 Jun 2023

PONE-D-21-26313R2 

Diagnostic accuracy of the Copenhagen Index in ovarian malignancy: a meta-analysis 

Dear Dr. Ma:

I'm pleased to inform you that your manuscript has been deemed suitable for publication in PLOS ONE. Congratulations! Your manuscript is now with our production department. 

Kind regards, 

on behalf of

Dr Federico Ferrari 

Academic Editor

PLOS ONE